# Anxious Brains: A Combined Data Fusion Machine Learning Approach to Predict Trait Anxiety from Morphometric Features

**DOI:** 10.3390/s23020610

**Published:** 2023-01-05

**Authors:** Teresa Baggio, Alessandro Grecucci, Federica Meconi, Irene Messina

**Affiliations:** 1Clinical and Affective Neuroscience Lab (CLI.A.N. Lab), Department of Psychology and Cognitive Sciences (DiPSCo), University of Trento, 38068 Rovereto, Italy; 2Centre for Medical Sciences, CISMed, University of Trento, 38122 Trento, Italy; 3Department of Economics, Universitas Mercatorum, 00186 Rome, Italy

**Keywords:** trait anxiety, machine learning, independent component analysis, structural networks, decision tree, emotion regulation

## Abstract

Trait anxiety relates to the steady propensity to experience and report negative emotions and thoughts such as fear and worries across different situations, along with a stable perception of the environment as characterized by threatening stimuli. Previous studies have tried to investigate neuroanatomical features related to anxiety mostly using univariate analyses and thus giving rise to contrasting results. The aim of this study is to build a predictive model of individual differences in trait anxiety from brain morphometric features, by taking advantage of a combined data fusion machine learning approach to allow generalization to new cases. Additionally, we aimed to perform a network analysis to test the hypothesis that anxiety-related networks have a central role in modulating other networks not strictly associated with anxiety. Finally, we wanted to test the hypothesis that trait anxiety was associated with specific cognitive emotion regulation strategies, and whether anxiety may decrease with ageing. Structural brain images of 158 participants were first decomposed into independent covarying gray and white matter networks with a data fusion unsupervised machine learning approach (Parallel ICA). Then, supervised machine learning (decision tree) and backward regression were used to extract and test the generalizability of a predictive model of trait anxiety. Two covarying gray and white matter independent networks successfully predicted trait anxiety. The first network included mainly parietal and temporal regions such as the postcentral gyrus, the precuneus, and the middle and superior temporal gyrus, while the second network included frontal and parietal regions such as the superior and middle temporal gyrus, the anterior cingulate, and the precuneus. We also found that trait anxiety was positively associated with catastrophizing, rumination, other- and self-blame, and negatively associated with positive refocusing and reappraisal. Moreover, trait anxiety was negatively associated with age. This paper provides new insights regarding the prediction of individual differences in trait anxiety from brain and psychological features and can pave the way for future diagnostic predictive models of anxiety.

## 1. Introduction

Anxiety refers to a condition of intense apprehension, tension, and worry complemented by the activation of the autonomic nervous system that causes a set of typical physiological reactions including increased heart rate and blood pressure, sweating, nausea, dizziness, hyperventilation, and muscle tension [1]. Considering that 45.82 million diagnoses of anxiety disorders were reported globally in 2019 [2], it is clear that there is an urgency to understand anxiety in order to improve diagnostic prediction and available treatment possibilities [3,4,5].

According to a long tradition tracing back to the conceptual elaborations and theories of Freud [6], Cattel, and Scheier [7], anxiety can be considered both a state, namely a transient emotion characterized by negative feelings (such as apprehension and tension) and physiological arousal, and a trait, namely when it refers to a more-stable individual’s predisposition to constantly perceive such negative feelings [8,9]. State anxiety and trait anxiety have been object of neuroimaging investigations in order to identify their structural and functional brain correlates. State anxiety has been investigated through the observation of brain responses during the exposure of anxiety-eliciting stimuli in animals and humans, leading to the definition of a neurobiological model that views anxiety as the result of an interaction between emotional reactivity and emotional regulation processes, in which an initial evaluation is carried out by a set of limbic structures and a successive evaluation is carried out by cortical areas [10]. The limbic core includes the amygdala, the insula, and interconnected structures such as the periaqueductal gray (PAG) and the hypothalamus. The cortical areas that participate to the successive evaluation include the anterior cingulate cortex (ACC) and the lateral prefrontal cortex. The ACC, along with the lateral prefrontal cortex, have been suggested to mediate the top-down control over threat-related stimuli. For what concerns trait anxiety, neuroimaging studies have shown that it is associated with a variation in gray matter volume of the hypothalamus, the left thalamus [11], the amygdala, the inferior temporal gyrus, the parahippocampal gyrus, the inferior frontal cortex [12] with abnormalities in the thickness of the ACC, the orbitofrontal cortex (OFC), the insula, and the temporal cortex [13]. Recent advances on this topic were made by Saviola and colleagues [5] who suggested that trait anxiety, being a more stable personality feature, predominantly impacts on anatomical features such as gray matter concentration. This study indeed showed a relation between trait anxiety and portions of the precuneus, the cuneus, the middle temporal gyrus and the cerebellum, suggesting a structural covariance with the default mode network (DMN). Nevertheless, some functional alterations have been associated with trait anxiety as well. A study by Modi and colleagues [14] reported a reduced functional connectivity of the DMN in subjects with high trait anxiety. Concurrently, studies have also suggested the presence of alterations in the functional integrity of other macro-networks, such as the executive control network (ECN) [15,16,17] and the salience network [17,18,19] in anxiety disorders.

Beside functional and gray matter alterations, white matter variations have also been reported in relation to trait anxiety. For example, some studies have found an association between trait anxiety and the structural integrity of an amygdala-ventromedial prefrontal cortex pathway [20], the uncinate fasciculus [21,22], and tracts in the left temporal lobe [23].

According to the neurobiological model of anxiety described above, structural alterations associated with trait anxiety can be related to a higher reactivity to anxiety-eliciting stimuli, whereas functional alterations of cortical areas have been interpreted as the neural correlate of emotion regulation deficits in anxious individuals [24].

However, previous studies suffered from some limitations. First, they have used mass-univariate statistical techniques that treat each voxel in isolation without taking into account statistical dependencies among voxels [25,26,27,28]. Also, individual differences were not taken into consideration, as only the average of individuals has been considered (e.g., high vs. low trait groups) [26,29]. In some cases, region of interest (ROI) analyses have been performed instead of whole-brain approaches, thus limiting results to a small set of a priori-defined regions [25,26,28]. Lastly, results from previous studies present limitations with respect to the sample considered, as they were not tested for their generalization to new unobserved cases. Predicting new cases is a benchmark for using neuroscientific results in a future translational perspective [29].

In recent times, affective neuroscience has taken advantage from the application of multi-variate machine learning approaches (also named multi-voxel pattern analysis in the context of neuroscientific research) which allow researchers to test how distributed patterns across multiple voxels relate to experimental variables [29]. Compared to traditional univariate analyses, machine learning approaches have the benefit of being multivariate in nature as they perform a joint analysis of multiple voxels, while univariate approaches carry out separate analyses for each individual voxel; moreover, the generalization of results is not assumed, but empirically tested [29,30]. Based on the considerations stated above, the main purpose of this paper is to provide new fresh evidence on the neural bases of trait anxiety by using a combination of unsupervised and supervised machine learning approaches applied to gray and white matter features in a data fusion perspective. The aim is not only to explore the neural bases of anxiety, but also to extract a neural model to predict trait anxiety. As such, this model can pave the way for the creation of a potential biomarker to predict new cases [29,31].

The unsupervised machine learning approach used in this context consisted in the application of Parallel ICA [32,33], which is a modified independent component analysis (ICA) approach that is able to combine two modalities (gray and white matter) at the same time, assessing the relationship among them and decomposing the brain into naturally grouping gray and white matter networks with lower dimensionality [34]. This approach is also coherent with a network perspective in neuroscience [35], being built upon the notion that emotional processes are distributed across a subset of regions constituting a network. In our study, we decided to use gray and white matter because they can be informative and also subject to similar genetic influences [36]. Pathological processes are indeed not limited to gray matter, but they also extend to white matter [25,36]. Of note, white matter studies are usually based on diffusion tensor imaging (DTI), a technique that assesses the integrity of white matter fibers through the fractional anisotropy index, which measures the displacement of water molecules on the micron scale [37,38]. However, DTI is highly sensitive to noise and, because of this, the region-of-interest-driven protocols are often poorly reproducible [39]. The fact of considering the pure white matter concentration (similarly to gray matter) from a classical T1 image has the advantage to evaluate distributed white matter alterations without the constraints imposed by specific tracts.

After decomposing the brain into meaningful circuits, we aimed to use decision tree tegression [40], a supervised machine learning approach, to build and test a predictive model of trait anxiety based on morphometric features. Moreover, we aimed to use network analysis to better characterize the role of the brain network predicting trait anxiety in the context of the other brain networks decomposed by Parallel ICA. Previous clinical and experimental observations [41] have reported that anxiety interferes with highly anxious individuals’ cognitive abilities, leading to increased negative thinking, augmented memory, and attentive performances for anxiety-related stimuli, and even to potentiated motor reflexes [42,43]. We expected to find that anxiety-related networks have a central role in modulating other networks not primarily related with anxiety. For this reason, we predicted that the anxiety-related networks have a high degree (the measure of the total number of links/edges that the node has) and expected influence over the other brain networks.

In the present study, we are also interested in testing the hypothesis that the usage of specific cognitive emotion regulation strategies could be associated with trait anxiety. Cognitive emotion regulation strategies refer to strategies that are used to alter “individual’s thoughts after having experienced a negative event” [44]. The literature has extensively reported an association between anxiety disorders and maladaptive cognitive strategies such as rumination [45,46,47,48] and catastrophizing; however, the role of the other strategies in predicting anxiety still needs further investigation. In this respect, we used the scales of the cognitive emotion regulation questionnaire (CERQ) [49] to identify the role of four maladaptive strategies (self-blame, rumination, catastrophizing, and other-blame) and five adaptive strategies (acceptance, positive refocusing, refocus on planning, putting into perspectives, and positive reappraisal). We expected to find the negative strategies to be highly associated with trait anxiety, especially rumination and catastrophizing, as they characterize the pathological forms of anxiety [49]. We also expected some of the positive strategies identified by the CERQ, such as reappraisal and refocusing, to be negatively related to trait anxiety. Previous studies have suggested that individuals with high anxiety do not rely on positive emotion regulation strategies [50].

Additionally, we were also interested in testing the hypothesis that trait anxiety may change with ageing. Previous research has reported mild evidence that generalized anxiety may decrease with age [51,52,53]. Beside generalized anxiety, we predicted similar results (negative relationship) for trait anxiety.

## 2. Materials and Methods

### 2.1. Participants

Behavioral and structural MRI data from 160 participants (M = 104, F = 56, age range = 20–80, mean age range = 39.28) were selected from the Leipzig mind-body-brain interactions (LEMON) protocol [54] included in the general MPI-Leipzig mind-brain-body dataset (OpenNeuro Dataset, https://openneuro.org (accessed on 1 December 2021), accession number ds000221 [55]), which consisted of structural, functional and behavioral data of 318 subjects. Participants of the MPI-Leipzig mind-brain-body dataset were selected according to specific criteria, which included medical eligibility for magnetic resonance sessions and absence of past or present psychiatric and neurological disorders. Additional exclusion criteria were applied during our subject’s selection, such as the intake of medication, positivity to the drug screening, present or past diagnosis of mental disorders apart from those related to anxiety, present or past alcohol abuse, and left handedness.

The data collection was authorized by the ethics committee of the University of Leipzig (154/13-ff) [54].

Two subjects were excluded for artifacts leading to a final sample of 158 subjects (104 males and 56 females; Mean age range = 39.28; SD = 20.26).

### 2.2. Questionnaires

The German version of the State-Trait Anxiety Inventory (STAI-G-X2) [8,9] was used to predict anxiety traits and consisted of 20 items, with a 4-point Likert scale ranging from 1 (almost never) to 4 (nearly always). Participants obtained a mean score of 36/±8.47. STAI-T assesses the stable propensity of individuals to perceive stressful situations as threatening and to respond to them with an increase in anxiety-related symptomatology [8].

The cognitive emotion regulation questionnaire (CERQ) [44,56] was taken into consideration to assess the usage of emotion regulation strategies. The CERQ included nine scales, which measured adaptive (acceptance, positive refocusing, refocusing on planning, positive reappraisal, and putting into perspective) and maladaptive (self-blame, rumination, catastrophising, and other-blame) emotion regulation strategies, rated on a 5-point Likert scale ranging from 0 (almost never) to 5 (almost always).

### 2.3. MRI Data

Quantitative T1-weighted images were acquired at the day clinic for cognitive neurology of the University Clinic Leipzig and the Max Planck Institute for Human and Cognitive and Brain Sciences (MPI CBS) in Leipzig, Germany. Magnetic resonance imaging (MRI) was performed on a 3T Siemens MAGNETOM Verio scanner (Siemens Healthcare GmbH, Erlangen, Germany) with a 32-channel head coil. The original LEMON protocol comprised fMRI scan and diffusion-weighted imaging (DWI) scan [54], but for the purposes of our research, we only took into consideration the T1-weighted images. The MP2RAGE sequence consisted of the following parameters: sagittal acquisition orientation, one 3D volume with 176 slices, TR = 5000 ms, TE = 2.92 ms, TI1 = 700 ms, TI2 = 2500 ms, FA1 = 4°, FA2 = 5°, pre-scan normalization, echo spacing = 6.9 ms, bandwidth = 240 Hz/pixel, FOV = 256 mm, voxel size = 1 mm isotropic, GRAPPA acceleration factor 3, slice order = interleaved, duration = 8 min 22 s.

### 2.4. Preprocessing

T1-weighted images were pre-processed through SPM12 (Statistical Parametric Mapping, https://www.fil.ion.ucl.ac.uk/ (accessed on 10 December 2021)) [57] and CAT12 toolbox (Computational Anatomy Toolbox for SPM, http://www.neuro.uni-jena.de/cat/ (accessed on 10 December 2021)) [58]. First, we manually re-oriented all the images placing the anterior commissure as the origin. After that, we computed the segmentation into gray matter, white matter and cerebrospinal fluid. The registration was computed through Diffeomorphic Anatomical Registration using Exponential Lie algebra tools for SPM12 (DARTEL) [59]. Finally, we performed the normalization to the MNI space with a spatial Gaussian smoothing of 10. During the procedure, data from two subjects encountered processing errors, and this led to the elimination of two participants. As a result, the final sample resulted in 158 subjects.

### 2.5. Data Fusion Unsupervised Machine Learning to Decompose the Brain into Independent Networks

For the network decomposition, Parallel ICA was applied to structural data using the Fusion ICA Toolbox (FIT, http://mialab.mrn.org/software/fit (accessed on 1 January 2022)) [60] in the MATLAB 2021b environment (https://it.mathworks.com/products/matlab.html (accessed on 1 January 2022)) [61].

The number of components was estimated for both modalities with information theoretic criteria [62]. To assess the consistency of each modality, ICASSO [63,64] was run ten times and the Infomax algorithm was selected. The resulting output consists in a matrix with the number of subjects (rows) and the loading coefficients for each component (columns). Loading coefficients represent how each component is expressed for every subject. As a final step, we converted the independent components into Talairach coordinates in order to explicit the brain areas comprising them. Areas with both positive and negative values, if present, were considered and plotted in Surf Ice (https://www.nitrc.org/projects/surfice/ (accessed on 1 January 2022)), using a different template for gray and white matter.

### 2.6. Supervised Machine Learning to Predict Anxiety Traits

The loading coefficients of each network, for each participant, were then entered into a supervised machine learning system to predict anxiety scores. To avoid redundancy and collinearity problems in the features considered, only the gray matter coefficients of every circuit were considered. A decision tree regression model was used in JASP [65]. A decision tree is a supervised machine learning algorithm that uses a decision tree as a model in order to obtain predictive estimates for the variables that take continuous or ordered discrete values [40]. The algorithm finds the optimal decision tree calculating the error between the predicted value and the actual value at each split point. The split point errors across the variables are then compared and the lowest prediction error is taken for the tree generation [66]. In the construction of the tree, each gray matter component was evaluated in the overall prediction process, with the calculation of the impurity reductions over the nodes. The hold-out method was used to test the prediction value of the model with the following split: 80% of observations were used to train the model, and 20% were left out to test the model (in other words, 127 observations were used for training the model and 31 observations for testing the model in its ability to predict new unobserved cases). The algorithmic settings were as follow: minimum observations for split were set to 20; minimum observations in terminal to 7; max iterations depth to 30; complexity parameter equal to 0.01. Predictors were scaled. Furthermore, the loading coefficients were entered into a backward regression in JASP. This was conducted to provide eventual converging evidence of the circuits predicting trait anxiety. Figure 1 shows a simplified representation of the combined unsupervised and supervised machine learning approach that was employed.

### 2.7. Network Analysis

To further explore the role of the anxiety-predicting components found in the previous steps, a network analysis was run.

Network analysis allows for a visual inspection of the interactions between a large numbers of variables, represented by the nodes. The nodes are connected to each other through the edges, and are placed according to the Fruchterman–Reingold algorithm [67] that organizes the network following the nodes’ strength of connection so that closer nodes have a stronger relationship [65]. To enhance the accuracy and the interpretability of the network, we chose EBICglasso [68,69] as estimator, with *γ* hyperparameter of 0.5, favoring the selection of models with fewer edges. The selected model evaluates the partial correlation between the variables, controlling for the effects of all the other measured variables in the network, meaning that if two nodes are connected together, the correlation between them cannot be explained by the other variables in the network [70].

Among the parameters computed by network analysis, we considered centrality measures, which specify the relevance of each node in the system. Such measures are betweenness, closeness and strength: betweenness provides information about the importance of a node in the average pathway between other pairs of nodes, closeness informs about the relationship of one node with all the others (with higher values indicating shorter distance between a specific node with the others), strength quantifies how strongly a node is connected to the other nodes, taking into account the sum of the weighted numbers, and strength of all connections of a specific node with all the others [70]. The evaluation of these measures allows for an interpretation of the nodes’ roles at the network level.

### 2.8. Questionnaire Analysis

A backward regression with STAI and CERQ scores was performed in order to understand which cognitive emotion regulation strategy predicts trait anxiety. Moreover, to find out whether anxiety decreases with age, a correlation was made between STAI and age. Both backward regression and correlation were performed inside JASP.

## 3. Results

### 3.1. Networks Decomposition

The information theoretic criteria estimated 13 independent covarying gray (IC-GM) and white (IC-WM) matter networks (see Appendix A). The positive values of these networks indicate increased gray/white matter concentration, whereas negative values indicate decreased concentration. The meaning of the covariation between a gray and a white matter component refers to a similar pattern of gray/white matter concentration.

### 3.2. A Predictive Model for Trait Anxiety

Decision tree regression returned a R^2^ of 0.271, MSE 0.596, RMSE 0.772, MAE 0.654, MAPE 1365.78%. A list of the gray matter components, ordered according to the relative importance (feature importance), can be found in Table 1.

Additional backward regression on the same ICs loading coefficients returned a similar significant model (R = 0.270, R^2^ = 0.073, Adjusted R^2^ = 0.055, RMSE = 8.282, *p* = 0.008). This model includes IC-GM 5 (*p* = 0.042) and IC-GM 11 (*p* = 0.022) as surviving predictors. This analysis further confirmed the results from decision tree regression, with the only exception of IC-GM4 which was not included in the backward regression results.

Following the converging evidence provided by the decision trees (that measures the generalizability of the model to new unobserved cases) and the backward regression (that measures the robustness of the results limited to the considered sample), we focused our attention on the two networks confirmed by both analyses: IC-GM5 (that covaries with IC-WM4) and IC-GM11 (that covaries with IC-WM8). See Table 2 for the anatomical denominations of each network’s areas, and Figure 2 and Figure 3 for a visual representation of them.

### 3.3. Network Analysis

Network analysis allowed us to evaluate the function of the anxiety-related components in the overall brain organization, checking their role in a wider systemic perspective. Figure 4 reports a visual representation of all the gray and white matter independent components (nodes) and their reciprocal connections (edges) at a systemic level.

Specific values for centrality measures are reported in Table 3. Regarding the centrality parameters, providing insight into the relative importance of a node with respect to the others [70], IC-GM5 and IC-GM11 show high levels of degree and expected influence and a small level of betweenness, while IC-WM8 shows a modest level of betweenness and degree but a moderate level of expected influence. IC-WM4 shows a higher number of connections with respect to IC-WM8, with a high level of strength and betweenness. Considering the centrality measures cited above and the network plot, it appears that network 1 (IC-GM5/IC-WM4) and network 2 (IC-GM11/IC-WM8) constitute two important hubs, along with network IC-GM3/IC-WM13, which also shows a great number of strong connections and a high expected influence.

### 3.4. Behavioural Analysis

Backward regression returned a significant model including six different emotion regulation cognitive strategies predicting anxiety traits (R = 0.578, R^2^ = 0.334, Adjusted R^2^ = 0.307, RMSE = 7.092). All the involved strategies reported a *p*-value < 0.05.

Catastrophizing, self-blame, other-blame, and rumination positively predicted trait anxiety scores, while positive refocusing and positive reappraisal were negatively associated to trait anxiety. Additional details are outlined in Table 4.

A negative significant correlation was also found between STAI and age (Pearson’s r = −0.250, *p*-value = 0.002). To further explore possible differences between young and old participants, we divided the subjects into two groups, according to the age: group 1 ranging from 20 to 49 years old (mean STAI = 37.5) and group 2 ranging from 50 to 80 years old (mean STAI = 33.2). A t-test implemented in JASP confirmed the significance in trait anxiety differences between the two groups (*p*-value = 0.002).

## 4. Discussion

The aim of the present study was to build a predictive model of individual differences in trait anxiety from gray and white matter features. Additionally, we were interested in testing the hypothesis that specific emotion regulation strategies could predict trait anxiety. Regarding the first point, a combined data fusion machine learning approach was used for the first time in this context. First, unsupervised machine learning (Parallel ICA) was used to decompose the brain into 13 different independent covarying gray-white matter networks. Then, a supervised machine learning (Decision tree regression) was used to build a predictive neural model of trait anxiety. Of note, this is the first study that combines unsupervised and supervised machine learning algorithms to understand trait anxiety. By combining UML and SML, we took advantages of both approaches. On one hand, UML such as pICA can provide a more biologically plausible way to decompose the brain into meaningful circuits that can outperform atlas-based parcellations, or a priori selected regions of interest. On the other hand, SML methods, such as a decision tree, can be used to build a predictive model of anxiety based on neural features (extracted via pICA). As such, Decision trees can be a valuable alternative to standard frequentist approaches in which generalization is assumed, but not empirically tested. Last but not least, we used a data fusion approach to take into account both gray and white matter contribution to a predictive model of trait anxiety.

By applying this combination of UML and SML, we found two networks that predicted trait anxiety: network1 (IC-GM5/IC-WM4) and network 2 (IC-GM11/IC-WM8). In the following sections these networks are discussed in detail.

### 4.1. A Parieto-Temporal Network Predicting Trait Anxiety

Network 1 included regions such as the postcentral and the precentral gyrus, the inferior parietal lobule, the precuneus, the cingulate gyrus, the middle temporal gyrus, and the anterior cingulate cortex.

Interestingly, some regions of this parieto-temporal network belong to the default mode network [71].

DMN is active during self-referential processing, future planning, external and internal cues evaluation, and emotion regulation [16,72], usually decreasing its activity during attentional or stimulus-dependent tasks [73]. Furthermore, the DMN has been found to be severely altered in anxiety disorders, possibly implying a disrupted interaction between focused attention and the subject’s emotional state [5,71,74]. Specific regions of the DMN, such as part of the anterior cingulate cortex, have been proposed to be implicated in emotion regulation strategies such as extinction and cognitive regulation, probably through the interaction with other networks (i.e., fronto-parietal) [16,75,76]. Thus, alterations in the functioning of this network may indicate an excessive allocation of attentional resources towards the external environment, as a way of detecting potential threats, with a deficit in the self-oriented processes [73,77].

Among the regions found in our results, the postcentral gyrus and the precentral gyrus have both been associated with state anxiety [5,78], while the inferior parietal lobe has been suggested to modulate sustained anxiety [79].

Moreover, recent evidence seems to suggest a role of the precuneus for rumination and anxiety [5] as well as for maintaining the sense of self and conscious information processing along with agency [80,81].

Network 1 also included white matter fibers passing through the medial frontal, postcentral, cingulate, and superior frontal gyrus. The middle longitudinal fasciculus connects the precuneus and the superior parietal lobule with the dorsolateral temporal pole and the superior temporal gyrus [81], feasibly being involved in precuneus information transmission. On the other hand, postcentral gyrus white matter abnormalities have been found in patients with generalized anxiety disorder [82].

The cingulum bundle is a huge white matter tract that connects frontal, parietal, and medio-temporal regions, comprising both long and short association fibers, transmitting principally to the cingulate gyrus, which is associated with multiple functions such as emotion, reward, conflict, and error detection [83]. Several studies have found an altered FA of the cingulum in disorders such as schizophrenia, depression, autism, obsessive-compulsive disorder, post-traumatic stress disorder, and panic disorder [84,85]. Cingulate cortex is also a central node of the DMN, and a recent study [86] indeed found a correlation between the mean FA value of the cingulum and the level of functional connectivity between regions belonging to the DMN (precuneus/posterior cingulate cortex and medial frontal cortex).

Finally, negative values (decreased concentration) were mainly related to the middle and superior temporal gyrus, which have been found to have a reduced volume in patients with different anxiety disorders [87,88,89,90], suggesting a potential role of these regions in the development or maintenance of anxiety [91].

### 4.2. A Fronto-Parietal Network Predicting Trait Anxiety

Network 2 included regions such as the superior, the medial and the middle frontal gyrus, the cingulate gyrus, the precuneus, and the insula.

Some portions of the middle cingulate gyrus and the precuneus are implicated in executive control functions, being part of the executive control network (also known as fronto-parietal network).

The ECN is involved in voluntary action control and cognitive conflict resolution, and research suggests that trait anxiety can be related to defects in the ECN functioning [92].

Individuals with high trait anxiety tend to show a decreased functional connectivity between some regions of the executive control network and regions implicated in conflicts detection, suggesting an impairment in cognitive processes that require attentional control [93].

Considering the other regions found in our results, the superior frontal gyrus has been involved in a variety of roles, according to its functional subdivisions. In particular, it is partially involved in some of the DMN functions, being connected to the mid and anterior cingulate cortex. It is also connected to the middle and inferior frontal gyri, which are involved in the executive control network, and to the precentral gyrus, the thalamus, and the frontal operculum, which are nodes of the motor control network [94].

The anterior cingulate cortex has been suggested to be a core structure for both cognitive and emotional processing [95] and gray matter alterations in the right anterior cingulate gyrus have been reported in different anxiety disorders [96].

Similarly, the insula has been found to be volumetrically altered in different anxiety disorders [97,98], being associated with unpleasant emotions and regulation of arousal [75,76]. Moreover, the insula has been found to be more active in anxiety-prone subjects during emotion processing [99].

White matter fibers of network 2 included the middle and the medial frontal gyrus, the cingulate gyrus, and the superior frontal gyrus.

The middle frontal gyrus subserves bottom-up attention processes [100], while superior frontal gyrus white matter fibers might instead facilitate higher order cognitive processes such as self-referential behaviors linked to the activity of the DMN and executive control functions, given its connections with the inferior frontal gyrus, the parietal lobule, the precuneus, and the parahippocampal gyrus [101].

In view of the evidence examined above in Section 4.1 and Section 4.2, we can hypothesize a primary relation of trait anxiety with brain areas which support emotion regulation and cognitive control, with a plausible presence of some macro-networks impairments, as suggested by previous studies [5,14].

Because of neuroplastic changes in gray and white matter, it would be interesting to develop specific trainings or interventions that could directly impact on such emotion regulation and cognitive control brain areas, in order to attenuate trait anxiety. Indeed, recently it has been demonstrated that is possible to reduce anxiety symptoms with a 6-week attention bias modification protocol, modifying anterior cingulate cortex gray matter volume and functional connectivity [102]. The implications of these neuroplastic changes could thus extend from the creation of specific exposure trainings to the design of definite noninvasive brain stimulation protocols.

### 4.3. Network Analysis

To further clarify the role of the components found for anxiety in a broader network perspective, we performed a network analysis. Betweenness, strength, and expected influence parameters were considered to evaluate the systemic role of trait anxiety-related networks. IC-WM4/IC-GM5 and IC-WM8/IC-GM11 networks showed high centrality parameters such as strength and expected influence. In other words, these networks are relevant hubs in the brain macro-organization. This notable result may indicate how higher levels of trait anxiety can modulate brain regions in other networks. It is in fact well-known, for clinicians and experimentalists, how high levels of anxiety exert a huge influence over almost all cognitive, perceptual, and motor functions (e.g., increased negative thinking, perceptual and attentive abilities, memory, startle reflex, etc.) [103].

Notably, white matter components resulted in having a significant importance in the overall network when considering betweenness and strength parameters.

### 4.4. Additional Analyses

Confirming but also expanding the results of previous research [104], we found that catastrophizing, self-blame, other-blame, and rumination cognitive emotion regulation strategies positively predicted trait anxiety scores, while positive refocusing and positive reappraisal showed the opposite trend. It has been suggested that cognitive emotion regulatory capacities have a role in the progress and maintenance of anxiety and thus it is important to consider them as critical mediators in the development of different emotional disorders [104].

Using a specific cognitive emotion regulation strategy when facing a negative event can indeed facilitate or worsen the mastering of future adverse life experiences. Rumination especially, which refers to the persistent focus on the feelings and thoughts associated with negative events and their consequences, can contribute to a difficulty in the production and implementation of effective solutions to problems, less propensity to engage in other constructive activities, and to the tendency to experience less social support [105]. Exaggerated rumination can lead to catastrophizing as well.

According to our statistical analysis, both self-blame and other-blame resulted in predicting trait anxiety scores. In the case of self-blaming, this could be related to the experience of an overstated sense of guilt, while in the case of other-blaming this could be associated to the sense of guilt chronic avoidance. Both the sense of guilt and the sense of guilt chronic avoidance may indeed generate anxiety.

Interestingly, in adults, the strategies that mostly relate to anxiety symptoms, through a positive or negative association, seem to be catastrophizing, positive reappraisal, rumination, and self-blame [106], while in clinically anxious adolescents positive reappraisal tends to be less used [107].

These findings indicate that it may be useful to consider possible interventions at the level of the cognitive emotion regulation strategies, given that maladaptive thoughts and believes can highly impact symptomatology at the individual level [28,108,109,110].

Finally, as expected, a negative correlation between trait anxiety and age was found, suggesting that as age increases, there is a decrease in the reported anxiety. The reasons behind this could imply a possible reduction in the responsiveness for negative emotions and an increased emotional control (learned with the experience) [52].

## 5. Conclusions and Limitations

Our study successfully found a predictive model of trait anxiety from brain gray and white matter features by using an innovative combination of supervised and unsupervised machine learning approaches and data fusion.

The results revealed that a parieto-temporal and a fronto-parietal structural network predict individual differences in trait anxiety. Moreover, additional analyses highlighted the positive association of trait anxiety with catastrophizing, rumination, other- and self-blame, a negative correlation instead with positive refocusing and reappraisal, and a negative trend associated to age.

Our study does not come without limitations. Firstly, we have used only structural brain features. Future studies may want to explore a fusion between structural and functional MRI (resting state or task-related). Moreover, although we used a larger number of subjects compared to previous research on this topic, future studies with larger samples are needed in order to confirm our results.

That said, to our knowledge, this data fusion combined with machine learning approach has not been applied previously in the context of trait anxiety.

Behavioral analysis gave us an insight on the emotion regulation cognitive strategies that could generate and maintain anxiety states, specifically catastrophizing, rumination, self-blame, other-blame, positive reappraisal, and positive refocusing. Lastly, we confirmed the tendency of reported anxiety to decrease with age.

Our findings could be useful for the development of new diagnostic tools, such as specific neuroimaging biomarkers, and the use of neurostimulation-based methods for the treatment of anxiety, also in subclinical subjects, in order to attenuate its symptoms [5,109].

## Figures and Tables

**Figure 1 sensors-23-00610-f001:**
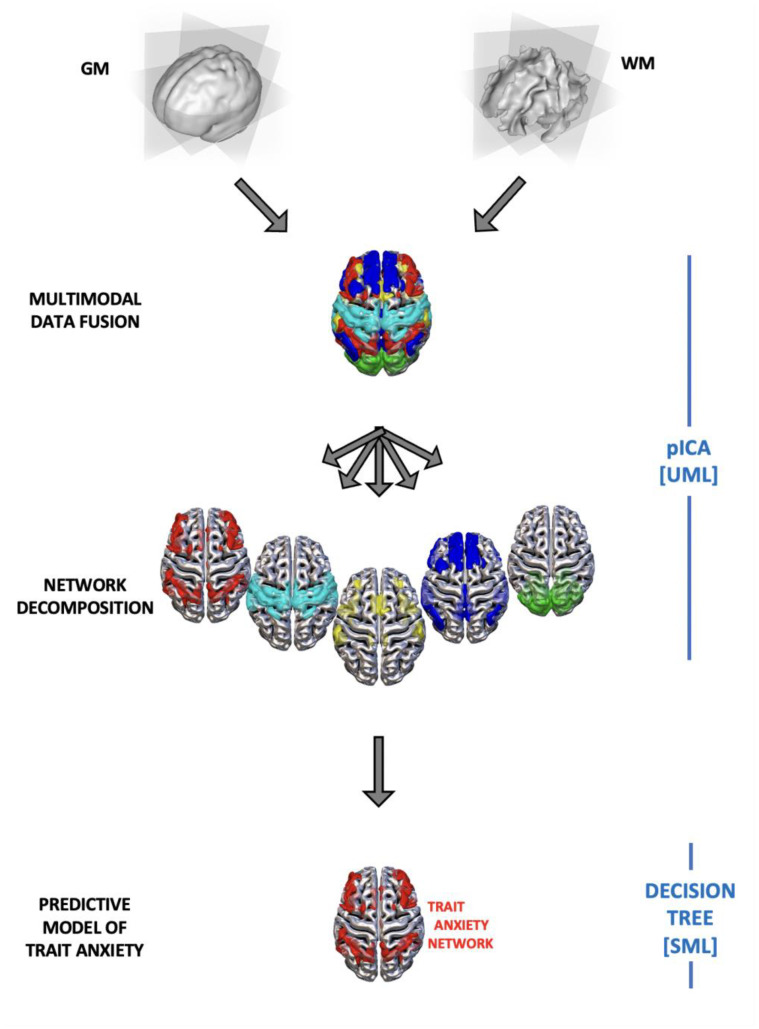
Combined data fusion machine learning approach. First unsupervised machine learning Parallel ICA (pICA) was used to combine the two modalities (gray and white matter) and to decompose the brain into the covarying gray-white matter independent networks. Then supervised machine learning decision tree (DT) was used to predict anxiety traits scores.

**Figure 2 sensors-23-00610-f002:**
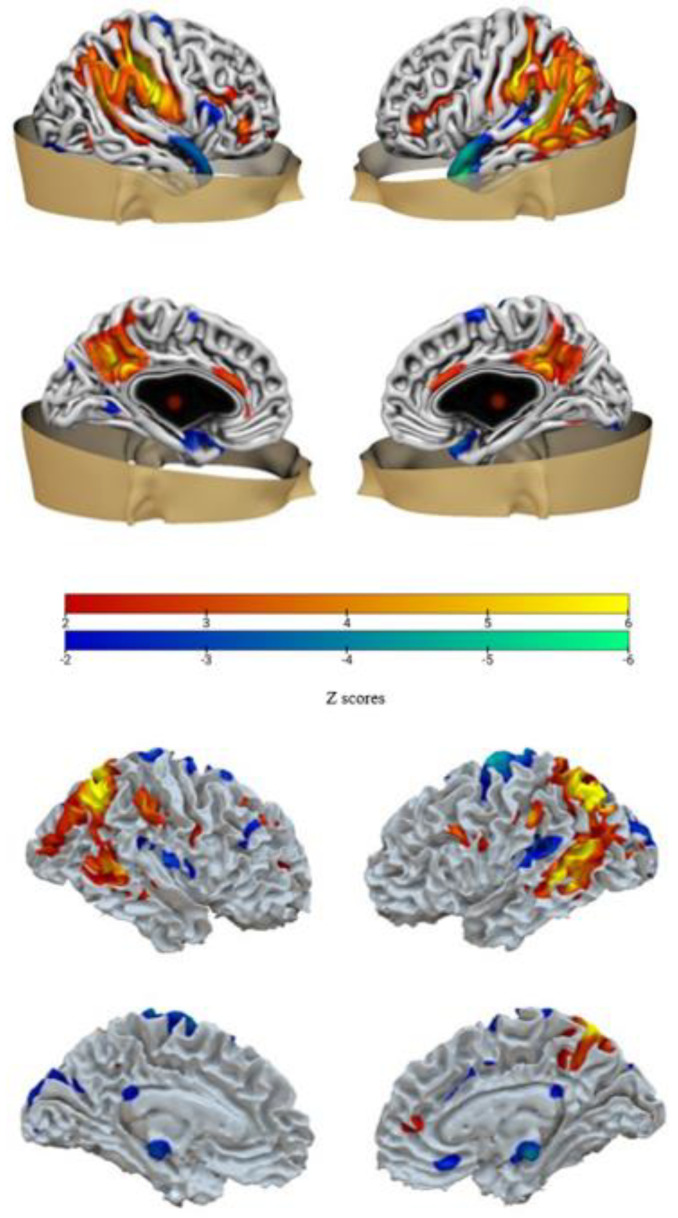
Network 1. Brain plot of IC-GM5 (**top**) and of IC-WM4 (**bottom**). Warm colors represent region with positive values, while cold colors represent regions with negative values.

**Figure 3 sensors-23-00610-f003:**
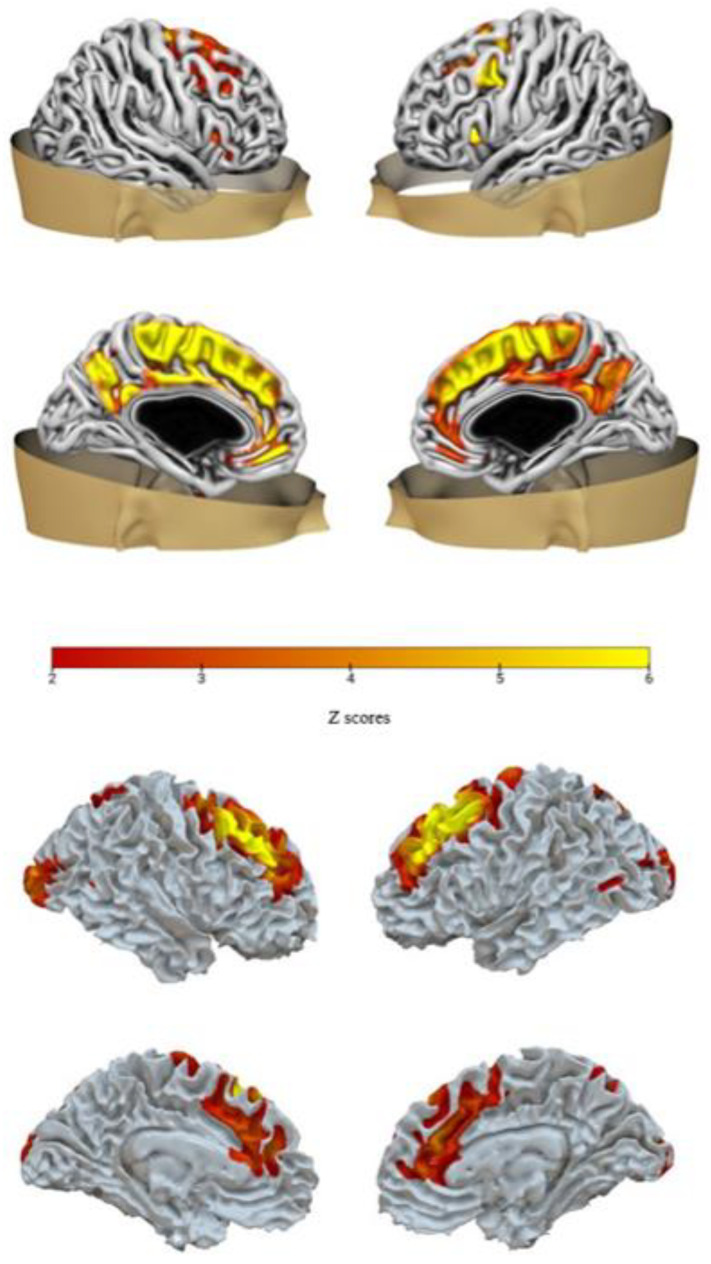
Network 2. Brain plot of IC-GM11 (**top**) and of IC-WM8 (**bottom**). Only positive values were found for both components.

**Figure 4 sensors-23-00610-f004:**
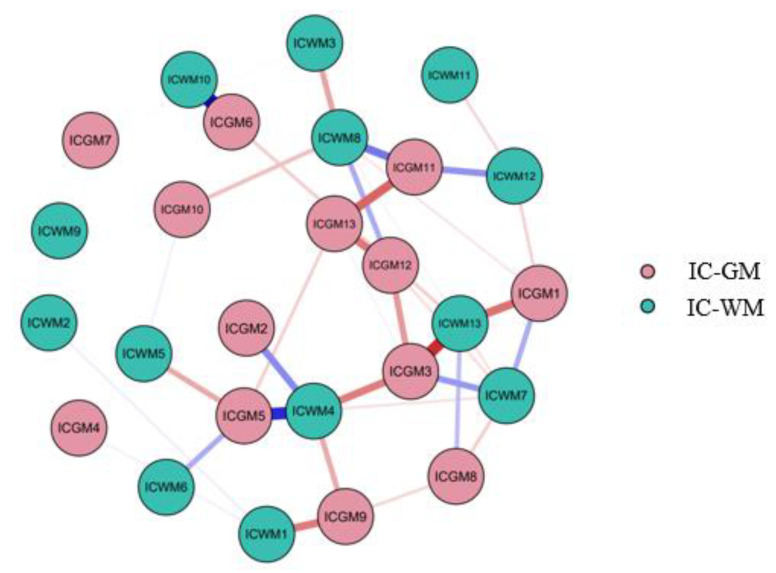
Network Analysis. Network plot of all components. Positive relationships are represented in blue, negative relationship are represented in red.

**Table 1 sensors-23-00610-t001:** Relative importance of the components in the decision tree Regression analysis.

Component	Relative Importance
ICGM11	13.467
ICGM4	11.155
ICGM5	10.620
ICGM1	9.230
ICGM2	8.916
ICGM8	7.700
ICGM9	7.466
ICGM13	6.628
ICGM3	6.533
ICGM7	6.072
ICGM12	5.650
ICGM10	5.382
ICGM6	1.183

Higher values of Relative Importance indicate higher influence of the component in the decision tree.

**Table 2 sensors-23-00610-t002:** Brain areas for each independent component, according to the Talairach daemon.

Independent Component IC-GM 5 Positive Values			
Area	Brodmann Area	Volume (cc)	MNI (x, y, z)
Postcentral Gyrus	1, 2, 3, 5, 40, 43	5.5/10.9	(−45, −30, 40)/(43, −27, 42)
Precentral Gyrus	4, 6, 43	1.6/4.9	(−52, −21, 37)/(49, −22, 39)
Superior Temporal Gyrus	13, 22, 39	4.7/3.0	(−43, −57, 22)/(43, −54, 22)
Sub-Gyral	10, 40	4.6/4.1	(−37, −31, 39)/(39, −33, 46)
Inferior Parietal Lobule	2, 7, 39, 40	9.3/8.8	(−40, −31, 42)/(39, −33, 40)
Middle Temporal Gyrus	21, 22, 37, 39	10.9/0.6	(−40, −60, 22)/(40, −58, 22)
Supramarginal Gyrus	40	2.9/3.3	(−40, −57, 25)/(45, −57, 25)
Precuneus	7, 19, 31	7.2/2.9	(−45, −51, 34)/(25, −63, 40)
Superior Parietal Lobule	5, 7	1.7/1.4	(−33, −73, 43)/(25, −64, 45)
Cingulate Gyrus	31	2.7/2.6	(−45, −46, 36)/(75, −45, 36)
Angular Gyrus	39	2.0/0.5	(−43, −66, 31)/(46, −61, 33)
Middle Occipital Gyrus	19, 37	1.9/0.1	(−42, −72, 3)/(51, −58, −75)
Cuneus	19	0.4/0.0	(−28, −82, 31)/(0, 0, 0)
Anterior Cingulate	24	0.4/0.1	(0, 28, 21)/(3, 30, 18)
**Independent component IC-GM 5 negative values**			
**Area**	**Brodmann Area**	**Volume (cc)**	**MNI (x, y, z)**
Superior Temporal Gyrus	22, 38	5.4/4.0	(−43, 13, −33)/(37, 18, −31)
Middle Temporal Gyrus	21, 38	3.3/0.2	(−43, 10, −36)/(63, −3, −4)
**Independent component IC-WM 4 positive values**			
**Area**	**Brodmann Area**	**Volume (cc)**	**MNI (x, y, z)**
Angular Gyrus	39	1.2/1.0	(−31, −61, 36)/(37, 58, 36)
Sub-Gyral	37, 40	14.0/12.0	(−30, −58, 33)/(34, −55, 36)
Precuneus	7, 19, 31, 39	9.2/9.8	(−31, −64, 39)/(19, −55, 45)
Superior Parietal Lobule	7	1.6/1.2	(−31, −64, 43)/(27, −54, 45)
Inferior Parietal Lobule	7, 39, 40	2.9/4.5	(−34, −61, 39)/(37, −55, 39)
Supramarginal Gyrus	40	1.1/2.7	(−36, −54, 34)/(37, −52, 36)
Middle Temporal Gyrus	19, 20, 21, 22, 37	9.0/5.8	(−49, −48, −1)/(36, −66, 27)
Superior Temporal Gyrus	13, 22, 39	1.4/1.3	(−42, −55, 7)/(48, −49, 13)
Cingulate Gyrus	31	1.9/1.7	(−18, −46, 28)/(18, −48, 28)
Postcentral Gyrus	2	1.1/2.1	(−51,−22, 25)/(55, −22, 31)
**Independent component IC-WM 4 negative values**			
**Area**	**Brodmann Area**	**Volume (cc)**	**MNI (x, y, z)**
Sub-Gyral	6	4.0/0.5	(−15, −4, 58)/(16, −24, 60)
Medial Frontal Gyrus	6, 8, 32	3.7/1.0	(−15, −7, 55)/(13, −24, 57)
Cingulate Gyrus	24, 31	1.3/0.1	(−15, −7, 49)/(18, 1, 49)
Superior Frontal Gyrus	6	1.5/0.2	(−13, 9, 55)/(16, 10, 55)
Superior Temporal Gyrus	22, 41, 42	1.8/1.0	(−40, −27, 7)/(55, −6, −4)
**Independent component IC-GM-11 positive values**			
**Area**	**Brodmann Area**	**Volume (cc)**	**MNI (x, y, z)**
Superior Frontal Gyrus	6, 8, 9	4.9/9.0	(−3, 7, 61)/(1, 10, 57)
Medial Frontal Gyrus	6, 8, 9, 10, 11, 25, 32	12.6/12.9	(−1, −1, 61)/(3, −3, 61)
Middle Frontal Gyrus	6, 8, 9, 47	7.7/4.8	(−40, 4, 42)/(19, −13, 64)
Cingulate Gyrus	23, 24, 31, 32	9.8/8.8	(−3, 24, 40)/(4, 21, 43)
Precuneus	7, 31	4.5/4.0	(0, −55, 36)/(3, −52, 36)
Paracentral Lobule	4, 5, 6, 31	3.4/2.4	(0, −15, 49)/(3, −12, 49)
Anterior Cingulate	10, 24, 25, 32	3.1/2.7	(−1, 36, 28)/(4, 36, 28)
Precentral Gyrus	6, 9	1.3/0.4	(−40, 1, 39)/(16, −19, 67)
Insula	13, 47	1.7/0.9	(−34, 19, 6)/(36, 22, 6)
Inferior Frontal Gyrus	9, 13, 45, 46, 47	4.0/3.0	(−37, 22, 6)/(36, 25, 3)
Posterior Cingulate	23, 29, 30, 31	1.0/1.2	(−1, −54, 22)/(4, −54, 22)
Sub-Gyral	6, 8	1.7/1.5	(−34, 25, 0)/(19, −10, 61)
**Independent component IC-WM 8 positive values**			
**Area**	**Brodmann Area**	**Volume (cc)**	**MNI (x, y, z)**
Sub-Gyral	8, 40	17.3/22.3	(−34, 34, 21)/(34, 34, 21)
Middle Frontal Gyrus	6, 8, 9, 10, 11, 46	7.7/11.5	(−36, 19, 33)/(37, 34, 24)
Medial Frontal Gyrus	6, 8, 9, 10	6.1/5.1	(−16, 30, 31)/(22, 34, 21)
Anterior Cingulate	10, 24, 32	3.9/4.7	(−16, 33, 28)/(22, 37, 18)
Cingulate Gyrus	24, 31, 32	2.7/2.2	(−13, 27, 31)/(18, 22, 34)
Superior Frontal Gyrus	6, 9, 10	2.6/2.7	(−18, 42, 24)/(28, 43, 12)
Cuneus	17, 18	0.7/1.3	(−16, −93, 6)/(21, −91, 1)
Precuneus	7, 31	2.3/0.4	(−16, −63, 37)/(12, −58, 34)
Middle Occipital Gyrus	18, 19	0.0/1.2	(0, 0, 0)/(25, −8.5, −1)

Note. Positive values are related to an increased gray or white matter concentration, while negative areas are related to a decreased concentration. White matter regions have to be interpreted as regions of white matter tracts that pass nearby and across the resulted named areas.

**Table 3 sensors-23-00610-t003:** Centrality measures for all the components in the network analysis.

Variable	Betweenness	Strength	Expected Influence
ICGM1	−0.506	0.327	−0.610
ICGM2	−0.732	−0.666	0.951
ICGM3	2.939	1.792	−1.853
ICGM4	−0.732	−1.237	0.305
**ICGM5**	**0.265**	**1.267**	**1.068**
ICGM6	−0.211	0.370	1.521
ICGM7	−0.732	−1.299	0.234
ICGM8	−0.732	−0.450	0.058
ICGM9	0.696	0.122	−1.423
ICGM10	−0.732	−0.881	−0.083
**ICGM11**	**0.084**	**0.912**	**0.795**
ICGM12	1.579	0.663	−0.955
ICGM13	0.129	0.625	−1.834
ICWM1	0.288	−0.394	−0.412
ICWM2	−0.732	−1.217	0.328
ICWM3	−0.211	−0.786	−0.347
**ICWM4**	**2.214**	**1.855**	**0.818**
ICWM5	−0.732	−0.748	−0.234
ICWM6	−0.732	−0.849	0.743
ICWM7	−0.732	0.703	0.267
**ICWM8**	**0.718**	**1.003**	**0.383**
ICWM9	−0.732	−1.321	0.209
ICWM10	−0.732	0.104	1.822
ICWM11	−0.732	−1.102	−0.039
ICWM12	−0.188	−0.247	0.373
ICWM13	0.990	1.454	−2.086

Centrality measures reflect the importance of the nodes/components. The components of interest in our analysis are highlighted in bold.

**Table 4 sensors-23-00610-t004:** Model summary for STAI-T and CERQ subscales.

Coefficients					
Model	Unstandardized	Standard Error	Standardized	t	*p*
(Intercept)	32.503	2.034		15.981	<0.001
Catastrophizing	0.761	0.303	0.195	2.513	0.013
Positive Refocusing	−0.481	0.223	−0.158	−2.157	0.033
Self-Blame	0.689	0.280	0.179	2.458	0.015
Other-Blame	0.973	0.337	0.216	2.891	0.004
Positive Reappraisal	−0.659	0.232	−0.216	−2.840	0.005
Rumination	0.570	0.249	0.184	2.292	0.023

Note. The following covariates were considered but not included: Acceptance, Refocus On Planning, Putting Into Perspectives.

## Data Availability

The dataset analyzed during the current study is available in the MPI-Leipzig_Mind-Brain-Body repository, https://openneuro.org/datasets/ds000221/versions/1.0.0 (accessed on 1 December 2021). The complete LEMON Data can be accessed via Gesellschaft für wissenschaftliche Datenverarbeitung mbH Göttingen (GWDG) https://www.gwdg.de/ (accessed on 1 December 2021). Raw and preprocessed data at this location is accessible through web browser https://ftp.gwdg.de/pub/misc/MPI-Leipzig_Mind-Brain-Body-LEMON/ (accessed on 1 December 2021) and a fast FTP connection (ftp://ftp.gwdg.de/pub/misc/MPI-Leipzig_Mind-Brain-Body-LEMON/ (accessed on 1 December 2021)).

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
