# Peer review of "Anxious Brains: A Combined Data Fusion Machine Learning Approach to Predict Trait Anxiety from Morphometric Features"

_sensors, 2023, doi:10.3390/s23020610_

Round 1
Reviewer 1 Report
The studies described in the manuscript "Anxious brains: A combined data fusion machine learning approach to predict trait anxiety from morphometric features" by Baggio et al. performed a Network analysis to test the hypothesis that anxiety-related networks have a central role in modulating other networks not strictly associated with anxiety. The following are some suggestions/concerns:
1. Anxiety disorders include diagnoses of panic disorder, agoraphobia, post-traumatic stress disorder (PTSD), social anxiety disorder (social phobia), specific phobias, generalized anxiety disorder (GAD), and obsessive-compulsive disorder (OCT)). Were the patients included in the study classified according to these conditions?
2. This is a really nice study and I think the authors should also discuss the previous reports suggesting changes in the specific brain structure and functions following various anxiety treatments (Biol Psychiatry 2007 Nov 15;62(10):1119-25; Biol Psychol 2022 Jul;172:108353.
3. Previously Inter-individual trait anxiety has been shown to be related to the intrinsic brain activity in the vmPFC and dACC/aMCC (Neuroimage. 2016 Jun;133:408-416). Have the authors seen any association between different brain area networks and the extent of anxiety symptoms?
4. Further, the sex difference in the association between trait anxiety and white matter tract integrity has been reported previously (Neuroscience 2012 Aug 16;217:77-83). Were there any sex differences in the network connectivity in the current study?
5. Previous studies have observed increased trait anxiety with age and that older adults often experience more difficulties with attention, verbal memory, and language abilities. A trend in change in trait anxiety from young adults to middle age and then old age (over 70s) has also been reported. Therefore, it will be exciting to present anxiety-age relationship data in the different age groups of patients.
Author Response
Dear reviewer, please find below our point-by-point response:
- Anxiety disorders include diagnoses of panic disorder, agoraphobia, post-traumatic stress disorder (PTSD), social anxiety disorder (social phobia), specific phobias, generalized anxiety disorder (GAD), and obsessive-compulsive disorder (OCT)). Were the patients included in the study classified according to these conditions?
R: We thank the reviewer for this observation. The present study did not included patients having received a psychiatric diagnosis, but included a few subjects that according to the SCID administered to participants before data collection, had in their life some anxiety problems such as panic disorder, agoraphobia, specific phobia, social phobia and obsessive-compulsive symptoms. Please note that in the present study were interested in anxiety trait and not in specific anxiety diagnoses. For this reason, we treated all subjects as spanning from low to high anxiety independently from having had specific anxiety problems.
- This is a really nice study and I think the authors should also discuss the previous reports suggesting changes in the specific brain structure and functions following various anxiety treatments (Biol Psychiatry 2007 Nov 15;62(10):1119-25; Biol Psychol 2022 Jul;172:108353).
R: We thank the reviewer for this comment. We have added now a brief discussion concerning previous studies reporting changes in gray/white matter in the paper (lines 465-476, Section 4.2).
- Previously Inter-individual trait anxiety has been shown to be related to the intrinsic brain activity in the vmPFC and dACC/aMCC (Neuroimage. 2016 Jun;133:408-416). Have the authors seen any association between different brain area networks and the extent of anxiety symptoms?
R: We thank the reviewer for letting us clarifying this. The approach used in the present paper (pICA and decision tree regression) allowed the identification of two gray and white matter networks associated to trait anxiety. Each of these networks comprises different brain regions, whose gray/white matter concentration can successfully predict individual differences in trait anxiety scores. The contribution of each brain area can be visually assessed in the figures, where the different colours show diverse z scores. In the discussion section we speculated about the possible functional roles of these regions in relation to anxiety, however, given the nature of our analysis and the aim of our study (focused on network-level decomposition), we did not investigate further the association between specific brain areas and anxiety level.
- Further, the sex difference in the association between trait anxiety and white matter tract integrity has been reported previously (Neuroscience 2012 Aug 16;217:77-83). Were there any sex differences in the network connectivity in the current study?
R: We thank the reviewer for pointing this out. We investigated sex differences in relationship to all the independent components resulted from pICA (both gray and white matter) through a logistic regression implemented in JASP. No significant results emerged for both gray and white matter components, with the exception of component GM2. However, since this component was not related to the trait anxiety networks, we did not report this specific information.
- Previous studies have observed increased trait anxiety with age and that older adults often experience more difficulties with attention, verbal memory, and language abilities. A trend in change in trait anxiety from young adults to middle age and then old age (over 70s) has also been reported. Therefore, it will be exciting to present anxiety-age relationship data in the different age groups of patients.
R: We thank the reviewer for this observation. Beside the correlation analysis, we have now added further information on age-related anxiety considering two subgroups: one of young participants (20-49 years old), and one of older participants (50-80 years old). Results indicate a significant difference between the two subgroups, confirming that trait anxiety seems to decrease in older people (see lines 371-375, Section 3.4).
Reviewer 2 Report
1. Authors need to check the format of references. In the text, reference numbers should be placed in square brackets [ ]. For example, lines 130, 353, etc.
2. It is suggested to include more descriptions of the Network analysis in Section 2.7.
3. In my opinion, the authors should highlight the study's key findings in the Conclusion section.
4. There are many single-sentence paragraphs in the current version of the manuscript. Authors may consider to rewrite the single-sentence paragraphs by combining the main idea with supporting arguments.
Author Response
Dear reviewer, please find below our point-by-point response:
- Authors need to check the format of references. In the text, reference numbers should be placed in square brackets [ ]. For example, lines 130, 353, etc.
R: We thank the reviewer for this suggestion. We have now corrected the reference format.
- It is suggested to include more descriptions of the Network analysis in Section 2.7.
R: We thank the reviewer for letting clarifying this. We added information in the Section 2.7, clarifying the meaning of the parameters that we used, in the context of Network analysis (see manuscript, Section 2.7).
- In my opinion, the authors should highlight the study's key findings in the Conclusion section.
R: Following reviewer suggestion we have added the key findings in the Conclusion section (see lines 517-521).
- There are many single-sentence paragraphs in the current version of the manuscript. Authors may consider to rewrite the single-sentence paragraphs by combining the main idea with supporting arguments.
We apologize for this. We have now revised some paragraphs, particularly in the Discussion section, reducing the amount of single sentences.